# Screening of Novel Bioactive Peptides from Goat Casein: In Silico to In Vitro Validation

**DOI:** 10.3390/ijms23052439

**Published:** 2022-02-23

**Authors:** Ezequiel R. Coscueta, Patrícia Batista, José Erick Galindo Gomes, Roberto da Silva, Maria Manuela Pintado

**Affiliations:** 1Centro de Biotecnologia e Química Fina (CBQF)-Laboratório Associado, Escola Superior de Biotecnologia, Universidade Católica Portuguesa, Rua Diogo Botelho 1327, 4169-005 Porto, Portugal; mpintado@ucp.pt; 2Human Neurobehavioral Laboratory/Research Centre for Human Development (HNL/CEDH), Universidade Católica Portuguesa, Rua Diogo Botelho 1327, 4169-005 Porto, Portugal; 3Instituto de Biociências, Letras e Ciências Exatas (IBILCE)-Rua Cristovão Colombo, Campus São José do Rio Preto, Universidade Estadual Paulista (UNESP), 2265, Jardim Nazareth, São José do Rio Preto 15054-000, SP, Brazil; erick.galindo.zoo@hotmail.com (J.E.G.G.); roberto.silva@unesp.br (R.d.S.); 4Laboratório de Microbiologia, Tecnologia Enzimática e Bioprodutos—Universidade Federal do Agreste de Pernambuco (UFAPE), Avenida Bom Pastor s/n, Boa Vista, Garanhuns 55292-270, PE, Brazil

**Keywords:** bioactive peptide, bioinformatics, in silico prediction, antihypertensive activity, ACE inhibition, antioxidant activity, ORAC

## Abstract

Food-derived bioactive peptides are of great interest to science and industry due to evolving drivers of food product innovation, including health and wellness. This study aims to draw attention through a critical study on how bioinformatics analysis is employed in the identification of bioactive peptides in the laboratory. An in silico analysis (PeptideRanker, BIOPEP, AHTpin, and mAHTPred) of a list of peptides from goat casein hydrolysate was performed to predict which sequences could potentially be bioactive. To validate the predictions, the in vitro antihypertensive potential of the five peptides with the highest potential was first measured. Then, for three of these, gastrointestinal digestion was simulated in vitro, followed by the analysis of the resulting ACE inhibitory activity as well as antioxidant capacity. We thus observed that the use of new computational biology technologies to predict peptide sequences is an important research tool, but they should not be used alone and complementarity with various in vitro and in vivo assays is essential.

## 1. Introduction

Peptide discovery, especially from different sources of food proteins, has an important interest in potential advancements in biology, chemistry, pharmacology, medicine, biotechnology, and the food industry [1,2]. Peptides (with lengths ranging from 2 to 50 amino acids) have crucial roles as antimicrobials, growth factors, hormones, biological messengers, and neurotransmitters [2]. Peptides can exert several bioactivities such as antiangiogenic, antibacterial, anticancer, anti-fungal, and others [2,3]. Therefore, there is an interest in researching their chemical and biological properties.

Naturally, many proteins or protein fragments (peptides) perform their biological functions in their native form; however, some peptides require changes to become bioactive. There are three different approaches for peptide generation: the action of proteolytic microorganisms, the digestive enzymes in the gastrointestinal tract, and the external hydrolysis with proteolytic enzymes [3,4]. External enzymatic hydrolysis is the most common method for the generation of bioactive peptides; however, the fermentation method has been considerably relevant to products such as milk [1].

Research on food-derived bioactive peptides is a hot topic, especially from cow’s milk and milk products, in their identification, characterization, and use [5,6]. With much lower coverage, goat milk has also been studied and characterized for its bioactive peptides, with beneficial properties such as antioxidant, ACE inhibitory peptides, anti-diabetic, and antimicrobial. However, more studies are needed to validate these health claims [6].

Considering the potential of bioactive peptides as novel therapeutics, nutraceuticals, and functional food ingredients, the discovery and prediction of novel bioactive peptides is an exciting area of research. Metabolomic, proteomic, and genomic screening of toxins and other natural product sources can identify bioactive peptides [7]. Advances in peptide screening and computational biology have come to support this area. The number of bioactive peptide databases covering a range of activities is at a promising seed stage. A good example is BIOPEP, a database of biologically active peptide sequences resulting in a tool for the evaluation of proteins as bioactive peptide precursors [5]. Tools such as this, and computational biology in general, are important for developing general predictors of bioactive peptides and identifying candidate peptides most likely to be bioactive. Isolating, identifying, and characterizing are critical; therefore, the development of new technologies to improve this is important. The use of machine learning to identify functional peptides and protein sequence data is a major advance and can improve the rapid and accurate detection of new biopeptides. Sequence-based in silico approaches can be used to select the best peptides before their synthesis and testing in laboratory experimentation, thus optimizing the design of therapeutic peptides.

Improvements in peptide screening and computational biology will continue to support peptide drug discovery. However, this new technology must be complemented with in vitro studies. This article presents a holistic perspective on recent advancements in silico peptide prediction and their relationship with in vitro assays focused on selected peptides from a goat casein hydrolysate. Figure 1 schematizes the approach to our idea.

## 2. Results

### 2.1. In Silico Prediction

An in silico analysis of a list of peptides from goat casein hydrolysate (from a study in progress) was performed to predict which sequences could potentially be bioactive. Table 1 shows the top 10 peptides analyzed with PeptideRanker [8]. PeptideRanker is a server for the prediction of bioactive peptides based on a novel N-to-1 neural network. This server returns the probability that the peptide will be bioactive. PeptideRanker was trained at a threshold of 0.5, i.e., any peptide predicted over a 0.5 threshold is labeled as bioactive. Table 1 shows that only 5 of the 10 sequences are potentially bioactive, the most likely being SWMHQPP.

After this first selection, the five potentially bioactive sequences were considered for analysis with another widely used tool for in silico prediction of bioactive peptides, BIOPEP [9] Table 2 shows all the potential bioactivities that each peptide encodes in its sequence. The table shows two of the parameters used to classify potential bioactivities in the BIOPEP database, parameter A (frequency of occurrence of bioactive fragments in a protein sequence) and B (potential biological activity). Among the five peptides only two potential bioactivities were repeated, angiotensin-I converting enzyme (ACE) inhibitor and dipeptidyl peptidase IV (DPP-IV) inhibitor, respectively antihypertensive and antidiabetic activities. Based on ACE inhibitor and DPP-IV inhibitor, the ranking was not the same as with PeptideRanker. Concerning B, the peptide with the highest potential to inhibit ACE was QSLVYPFTGPIPNSL (#4 for PeptideRanker), while the peptide SWMHQPP (#1 for PeptideRanker) was in second place with almost the same potential as the sequence YPYQGPIVL (#5 for PeptideRanker). Likewise, the latter peptide showed the highest potential to inhibit DPP-IV, with SWMHQPP coming in second place once again.

Following the BIOPEP analysis, we focused the analysis more on ACE inhibitory activity, for which we resorted to other novel tools of increasing use in predicting potentially antihypertensive peptides, AHTpin [10] and mAHTPred [11]. These are online servers that use machine learning trained to predict antihypertensive peptides from the reported peptides’ characteristics of all lengths. Once again, the ranking was different from the previous ones (Table 3 and Table 4). While again the change was not dramatic, it was for the peptide SWMHQPP, for which the AHTpin estimated the least antihypertensive capacity, with a very low value for what can be considered to have that potential.

Now, returning to the results obtained by BIOPEP, when the IC_50_ is removed from the equation of B (Equation (1)) and converted from µM to µg mL^−1^, the following values were obtained: QSLVYPFTGPIPNSL, 4.1 µg mL^−1^; SWMHQPP, 72.6 µg mL^−1^; YPYQGPIVL, 86.2 µg mL^−1^; HQPPQPL, 129.3 µg mL^−1^; MHQPPQPL, 150.1 µg mL^−1^. Ref. [12] reported some antihypertensive oligopeptides encoded in the human β-casein: VMP (IC_50_ 10.0 µg mL^−1^), TVYTKGRVMP (IC_50_ 43.7 µg mL^−1^), PAVVLP (IC_50_ 26.8 µg mL^−1^), LPQNILP (IC_50_ 36.5 µg mL^−1^), PHQIYP (IC_50_ 22.6 µg mL^−1^), NPPHQIYP (IC_50_ 35.7 µg mL^−1^), VLPYP (IC_50_ 21.1 µg mL^−1^), and VLPIP (IC_50_ 16.7 µg mL^−1^) [12]. Likewise, [13] reported antihypertensive oligopeptides encoded in various β-caseins: SLVYP (IC_50_ 23.7 µg mL^−1^), FAQTQSLVYP (IC_50_ 28.8 µg mL^−1^), and HPFAQTQQSLVYP (IC_50_ 36.1 µg mL^−1^) [13]. If we consider peptides with even greater antihypertensive potential, Tavares et al. (2011) reported whey-derived peptides with remarkable inhibitory effect: DAQSAPLRVY (IC_50_ 12.2 µg mL^−1^), DKVGINYW (IC_50_ 25.4 µg mL^−1^), KGYGGVSLPEW (IC_50_ 0.7 µg mL^−1^) [14]. Furthermore, we have previously reported peptide hydrolysates of soy protein, i.e., a mixture of peptides, with IC_50_ values lower than 60 µg mL^−1^ [15]. That is why, when analyzing the inhibitory capacity of a pure peptide, we consider that an IC_50_ value higher than 50 µg mL^−1^ is not significant compared to the benchmarks. In this sense, a priori, only the peptide QSLVYPFTGPIPNSL would be antihypertensive according to the IC_50_ estimated by BIOPEP.

### 2.2. In Vitro Analyses

#### 2.2.1. Antihypertensive Activity

To validate the in silico predictions, we measured the ability of the five peptides to inhibit ACE in vitro (iACE). Table 5 (as raw) shows the values of the bioactivities, which accounted for the low inhibitory capacity of the peptides, being SWMHQPP the only one that presented a considerable capacity (IC_50_ 223 µg mL^−1^). Even so, as already specified for a pure peptide which is considered antihypertensive, this level of inhibitory activity is low. For SWMHQPP, when comparing the observed value with the value that we calculated from BIOPEP, the observed IC_50_ was 3.1 times higher. In the case of QSLVYPFTGPIPNSL, which was predicted as the one with the highest antihypertensive potential by BIOPEP and AHTpin, and second by mAHTPred, the inhibitory activity was one of the lowest, practically nil.

#### 2.2.2. Simulated Gastrointestinal Digestion

Now, BIOPEP establishes the potential bioactivity based on the sequences encoded in the peptides. Thus, if these sequences were released through a lytic process, such as that which occurs in gastrointestinal digestion, the bioactivity could increase and approach that predicted. Thus, the three peptides of greatest interest from the in silico predictions (QSLVYPFTGPIPNSL, SWMHQPP, and YPYQGPIVL) were selected for in vitro simulated gastrointestinal digestion. We performed the digestions considering reaching a concentration of 50 µg mL^−1^ in the simulation of the intestinal compartment. For the three peptides, in addition to the antihypertensive potential, we analyzed the antioxidant potential by ORAC. Bioactivity analyses were performed with both the peptides before digestion and after digestion.

Before digestion, we found a moderate antioxidant capacity (Table 5), particularly for QSLVYPFTGPIPNSL and YPYQGPIVL [16,17]. In the case of SWMHQPP, the antioxidant capacity was considerable, being the highest of the three. After digestion of the three peptides, we observed a high increase in the antioxidant capacities of all peptides. The digest of QSLVYPFTGPIPNSL showed the highest antioxidant activity, followed by YPYQGPIVL, with SWMHQPP being below these two. Although the values between the digested peptides were different, they all demonstrated high bioactivity. This accounted for the release of antioxidant sequences encoded in each of the peptides. Concerning this, what was striking is that BIOPEP only estimated QSLVYPFTGPIPNSL with antioxidant potential. Although this peptide ended up being the one that showed the highest antioxidant activity after ingestion, the other two proved to be antioxidants both before and after digestion.

On the iACE activity, digestion had no positive effect. None of the three digested peptides showed any inhibitory activity, so the IC_50_ was clearly not reached under the conditions tested. This means that if there is any inhibitory activity, it is well above 50 µg mL^−1^.

## 3. Discussion

The current interest in the study of peptides has triggered a technological evolution, becoming a common practice to predict bioactive peptide sequences from parent proteins. These sequences can be released by lysis of the protein structure, as is the application of enzymatic hydrolysis [3,15]. In turn, peptides produced by proteolysis can still release other bioactive sequences in the enzymatic hydrolysis of the gastrointestinal digestive process [15]. Different results are reported in the literature for the residual antioxidant capacity of peptides and casein hydrolysates, after the in vitro-simulated gastrointestinal digestion, with an increase, reduction, or unchanged in this activity [18,19,20]. Contreras et al. (2013) observed a small increase in the antioxidant capacity of the AYFYPEL peptide, after the digestion process, from the activity of 3.216 ± 0.114 to 4.160 ± 0.623 µmol TE µmol^−1^ peptide [21]. However, for the other analyzed peptides (RYLGY and YQKFPQY), the antioxidant activity was reduced.

The most used bioinformatics tools to predict putative biologically active sequences are based on databases constructed from known structures, structural and physicochemical analyses that allow establishing different residue interactions that provide certain bioactivities [2,22,23]. However, the information behind these databases is still quite limited, both in terms of the spectrum of bioactive properties that each sequence can exert and the mechanisms that endow these structures with these properties. From this work, we could see that different tools predict differently, leading to different results. Furthermore, it is still unclear which tool may be the most reliable for the peptides discussed here. Although PeptideRanker predicted these peptides with bioactive potential, without specifying the properties, in the case of the antioxidant property they presented as interesting, the order of significance did not correspond to the predicted one. On the other hand, BIOPEP did not even predict most of the analyzed sequences as antioxidants, the only one for which it did was QSLVYPFTGPIPNSL. However, BIOPEP did predict these peptides as antihypertensives, just as we were able to achieve the same result using AHTpin and mAHTPred, considered the most reliable tools when involves in predicting antihypertensive activity [2]. However, this bioactivity was not observed in in vitro validation tests, which exposes the limitations that these tools still have.

This result does not mean that these bioinformatics tools are obsolete or unreliable, but that they still need to develop and grow. Likewise, these same tools have proven to be very timely in other works, with which the global utility in the search for therapeutic peptides cannot be denied [2,15]. To this end, it is very important to continue developing in vitro and in vivo assays to relate peptide primary and secondary structures to different biological properties, as well as to increase the spectrum of potential properties that peptides already reported can exert.

Although this work is an initial exploratory work, it still needs to be further expanded, which would require in vitro analysis of the DPP-IV inhibition property, as well as establishing the fragmentation of peptide sequences along the gastrointestinal tract.

## 4. Materials and Methods

### 4.1. Materials

Pepsin (800–1000 U/mg protein), pancreatin (4 × USP), α-amylase, angiotensin-I converting enzyme (peptidyl-di-peptidase A, EC 3.4.15.1, 5.1 U/mg), bile salts, Trolox (6-hydroxy-2,5,7,8-tetramethyl-chroman-2-carboxylic acid), and AAPH [2,2′-azobis (2-amidinopropane) dihydrochloride] were obtained from Sigma-Aldrich (St. Louis, MO, USA) and used without further purification. Fluorescein [3′,6′-dihydroxyspiro (isobenzofuran-1 [3H], 9′ [9H]-xanthen)-3-one] was purchased from Fisher Scientific (Hanover Park, IL USA). The tripeptide Abz-Gly-Phe(NO_2_)-Pro was obtained from Bachem Feinchemikalien (Bubendorf, Switzerland). Tris [tris (hydroxymethyl) aminomethane] was obtained from Honeywell Fluka (Charlotte, NC, USA). The peptides SWMHQPP, HQPPQPL, MHQPPQPL, QSLVYPFTGPIPNSL, and YPYQGPIVL were purchased from GenScript Biotech (Leiden, The Netherlands).

### 4.2. Methods

#### 4.2.1. Bioactivity Prediction

To predict in silico the bioactivity of each peptide, a ranking was first performed by PeptideRanker [8], which is a server for the prediction of bioactive peptides based on a novel N-to-1 neural network [5]. The first 5 peptides in the ranking were then analyzed using the online BIOPEP database [9], estimating the different potential bioactivities [22,24]. One of the important parameters that are estimated by BIOEPEP is the potential biological activity of a protein (*B*) [µM^−1^]:(1)$$B=\frac{\sum_{i=1}^{k}\frac{a_{i}}{EC_{50i}}}{N}$$
where *a_i_* is the number of repetitions of the *i*-th bioactive fragment in protein sequence; *EC*_50*i*_ is the concentration of *i*-th bioactive sequence corresponding to its half-maximal activity [µM] or half-maximal inhibition (IC_50_) in case of peptides with inhibitory activity; *k* is the number of different fragments with given activity; *N* is the number of amino acid residues.

In parallel, these same hundred peptides were analyzed about their antihypertensive potential using the online tools AHTpin [10,25] and mAHTPred [11,23]. For AHTpin, peptides with support vector machine (SVM) scores > 0.0 were considered as predicted with the antihypertensive property.

The analysis with all servers was performed on 14 December 2021. An update of the data obtained by BIOPEP was performed by entering the platform on 15 February 2022.

#### 4.2.2. ACE Inhibitory Activity

The ACE inhibitory activity was carried out using the fluorometric assay described by Coscueta, Brassesco, and Pintado (2021) [26]. The iACE of each sample (raw peptide and digested) was evaluated in duplicate and expressed as the concentration capable of inhibiting 50% of the enzymatic activity (IC_50_). To calculate the IC_50_ values, non-linear modeling was used, and the results were expressed as μg mL^−1^ to inhibit 50% of the enzymatic activity.

#### 4.2.3. Antioxidant Activity

The ORAC assay was performed according to Coscueta, Brassesco, and Pintado (2021) [26]. Each sample (raw peptide and digested) was analyzed in duplicate and the results were expressed in µmol TE by mg of peptide (µmol TE mg^−1^), after the calculation of Trolox concentration through regression curve equation.

#### 4.2.4. Simulated Gastrointestinal Digestion

Simulated gastrointestinal digestion was performed according to the method described by Madureira et al. (2011) [27]. Briefly, mouth digestion was mimicked by introducing the peptide sequence (2.5 mg/mL) into 15 mL of a 1 mM CaCl_2_ solution of 100 U mL^−1^ α-amylase, under constant stirring (200 rpm) for 2 min at 37 °C, simulating masticatory movements [28]. A 1 M NaHCO_3_ solution was used to adjust the pH of artificial saliva to 6.9. Subsequently, a pepsin solution, 25 mg/mL, was added with a ratio of 0.05 mL/mL of a sample at pH 2.0 (simulated gastric solution, stomach) following incubation for 2 h, 37 °C at 130 rpm orbital agitation for the gastric phase. Finally, intestinal digestion and absorption were simulated by adjusting pH values to 5 using a 1 M NaHCO_3_ solution and adding a solution of bile salts and pancreatin to the digest [27].

#### 4.2.5. Statistical Analysis

Statistical analysis was carried out with the aid of RStudio V 1.2.1335. The mean values from two replicates were analyzed statistically by analysis of variance followed by Tukey’s post hoc test. Separation of means was conducted by using the least significant difference at the 5% level of probability.

## Figures and Tables

**Figure 1 ijms-23-02439-f001:**
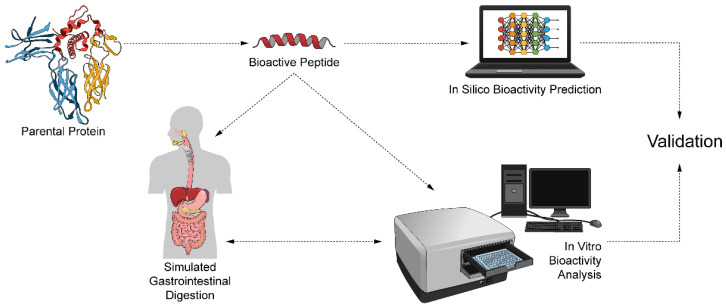
Process of analysis and validation of bioactive peptides performed in this article.

**Table 1 ijms-23-02439-t001:** Score peptides with the probability of performing biological activities, according to the PeptideRanker server.

Rank	Peptide Sequence	Score ^1^
1	SWMHQPP	0.846508
2	HQPPQPL	0.755043
3	MHQPPQPL	0.683869
4	QSLVYPFTGPIPNSL	0.612920
5	YPYQGPIVL	0.552820
6	WDQVKR	0.394225
7	YQEPVLGPVR	0.380921
8	PLTQTPVVVPPFLQP	0.277886
9	VMFPPQSVLS	0.267560
10	NFLKKISQ	0.147732

^1^ The score ranges from 0 to 1, and the closer to 1, the higher the probability of biological activity.

**Table 2 ijms-23-02439-t002:** Peptides with their potential bioactivities, and the corresponding calculations by BIOPEP.

Peptide	Activity	A ^1^	B ^2^
SWMHQPP	ACE inhibitor	0.4286	0.0017347
	Alpha-glucosidase inhibitor	0.1429	7.9 × 10^−6^
	DPP-III inhibitor	0.1429	
	DPP-IV inhibitor	0.7143	0.0006120
	Neuropeptide	0.1429	
HQPPQPL	ACE inhibitor	0.7143	0.0009012
	Alpha-glucosidase inhibitor	0.1429	7.9 × 10^−6^
	DPP-IV inhibitor	0.7143	2.44 × 10^−5^
MHQPPQPL	ACE inhibitor	0.6250	0.0007886
	Alpha-glucosidase inhibitor	0.1250	6.9 × 10^−6^
	DPP-IV inhibitor	0.7500	2.13 × 10^−5^
	Neuropeptide	0.1250	
QSLVYPFTGPIPNSL	ACE inhibitor	0.7333	0.0263997
	Antiamnestic	0.0667	
	Antioxidative	0.0667	
	Antithrombotic	0.0667	
	DPP-III inhibitor	0.1333	
	DPP-IV inhibitor	0.8000	0.0002435
	Hypotensive	0.0667	
	Opioid	0.1333	
	Regulating	0.0667	
	Stimulating	0.0667	
YPYQGPIVL	ACE inhibitor	0.4444	0.0013524
	Alpha-glucosidase inhibitor	0.2222	7.04 × 10^−5^
	Anti-inflammatory	0.1111	
	Antiamnestic	0.1111	
	Antithrombotic	0.1111	
	DPP-IV inhibitor	0.8889	0.0020040
	Regulating	0.1111	
	Stimulating	0.2222	

^1^ The frequency of bioactive fragments occurrence in protein sequence (A). ^2^ Potential biological activity (B) [µM^−1^].

**Table 3 ijms-23-02439-t003:** Prediction of potential antihypertensive activity of the peptides, according to the AHTpin web server.

Rank	Peptide Sequence	SVM Score ^1^
1	QSLVYPFTGPIPNSL	1.81
2	YPYQGPIVL	1.66
3	HQPPQPL	0.84
4	MHQPPQPL	0.83
5	SWMHQPP	0.31

^1^ The higher the support vector machine (SVM) score, the greater the antihypertensive potential.

**Table 4 ijms-23-02439-t004:** Prediction of potential antihypertensive activity of the peptides, according to the mAHTPred web server.

Rank	Peptide Sequence	AHTP or Non-AHTP ^1^	Probability
1	MHQPPQPL	AHTP	0.9816
2	QSLVYPFTGPIPNSL	AHTP	0.9773
3	HQPPQPL	AHTP	0.9773
4	YPYQGPIVL	AHTP	0.9570
5	SWMHQPP	AHTP	0.8748

^1^ Prediction of antihypertensive (AHTP) or non-antihypertensive (non-AHTP).

**Table 5 ijms-23-02439-t005:** In vitro bioactivities for pure raw and digested peptides.

Peptide Sequence	Raw		Digested	
	iACE ^1^	ORAC ^2^	iACE ^1^	ORAC ^2^
SWMHQPP	223 ± 12 *	2.64 ± 0.07 *^,†^	>50	16.83 ± 0.25 *^,†^
QSLVYPFTGPIPNSL	>3000	0.80 ± 0.03 *^,†^	>50	24.79 ± 0.10 *^,†^
YPYQGPIVL	1113 ± 42 *	1.54 ± 0.05 *^,†^	>50	20.94 ± 0.51 *^,†^
MHQPPQPL	>3000	n.a.	n.a.	n.a.
HQPPQPL	>3000	n.a.	n.a.	n.a.

^1^ Half-maximal inhibitory concentration (IC_50_) for ACE inhibitor activity measured as [µg mL^−1^], expressed as the mean ± SD ^2^ ORAC measured as [µmol TE mg^−1^ peptide], expressed as the mean ± SD * Values within the same column that present statistically significant differences. ^†^ Values from the same sample that, between treatments (raw and digested), present statistically significant differences. For conditions in which the analysis was not performed, it is designated as n.a. (not applicable).

## Data Availability

Data is contained within the article.

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
