# Peer review of "Screening of Novel Bioactive Peptides from Goat Casein: In Silico to In Vitro Validation"

_ijms, 2022, doi:10.3390/ijms23052439_

Round 1
Reviewer 1 Report
The authors used bioinformatics tools on a collection of peptides obtained from goat casein hydrolysate, to identify potentially bioactive sequences. They then validated their results experimentally.
I think their work is valid and of interest to the community. However, there are some minor concerns that need to be addressed before publication:
To verify the results, I ran the BIOPEP analysis for the 5 selected peptides. In the case of "SWMHQPP", I obtained different results from the ones reported in the manuscripts (Ace inhibitor A= 0.4286 instead of the reported A=0.2847). I also observed more activities than the ones reported in Table 2 for this particular peptide. The results I obtained for all other peptides were identical to the ones the authors reported. The results I obtained do not change the conclusions of the manuscript, but I would recommend updating the table before publication.
I wonder if the difference I observed is due to a mistake, or due to an update of the server which may lead to different results. In order to guarantee reproducibility, I would recommend reporting the date when the analysis was performed, or which version of the algorithm/server was used.
I also noticed that the website for AHTpin has been updated, and the new domain is the following: https://webs.iiitd.edu.in/raghava/ahtpin/index.php
I would recommend that for all servers, the date the results were obtained and versions of the servers are reported.
Overall, I think the kind of study reported here is valuable for the community. Bioinformatics results are valuable, but it is important that they are compared and verified experimentally.
Author Response
The authors used bioinformatics tools on a collection of peptides obtained from goat casein hydrolysate, to identify potentially bioactive sequences. They then validated their results experimentally.
I think their work is valid and of interest to the community. However, there are some minor concerns that need to be addressed before publication:
R: We appreciate your interest and constructive comments.
To verify the results, I ran the BIOPEP analysis for the 5 selected peptides. In the case of "SWMHQPP", I obtained different results from the ones reported in the manuscripts (Ace inhibitor A= 0.4286 instead of the reported A=0.2847). I also observed more activities than the ones reported in Table 2 for this particular peptide. The results I obtained for all other peptides were identical to the ones the authors reported. The results I obtained do not change the conclusions of the manuscript, but I would recommend updating the table before publication.
R: We appreciate the comment and the effort. We also corroborated the same, and even found that bioactivities were added for other peptides as well. The new results were updated in Table 2 as can be seen in track change.
I wonder if the difference I observed is due to a mistake, or due to an update of the server which may lead to different results. In order to guarantee reproducibility, I would recommend reporting the date when the analysis was performed, or which version of the algorithm/server was used.
R: The analyses in BIOPEP were performed some time ago. In the meantime, the complete platform has been updated. We followed the reviewer's suggestion and added the information on the analysis data.
I also noticed that the website for AHTpin has been updated, and the new domain is the following: https://webs.iiitd.edu.in/raghava/ahtpin/index.php
R: We appreciate the comment. At the time of article submission this update had not occurred.
I would recommend that for all servers, the date the results were obtained and versions of the servers are reported.
R: Thank you for your comment, we will add it to the text.
Overall, I think the kind of study reported here is valuable for the community. Bioinformatics results are valuable, but it is important that they are compared and verified experimentally.
R: We fully agree and that is why we are proposing it in the discussion.
Reviewer 2 Report
The paper by Coscueta and co-authors entitled “Screening of novel bioactive peptides from goat casein: in silico to in vitro validation” compares in silico results for bioactive peptides with in vitro analyses on antihypertensive and antioxidant activity. Although the topic of the manuscript is interesting it has not been well conducted. Extensive editing of English language and style is also required. In my opinion, the paper is not suitable for publication in the prestigious International Journal of Molecular Sciences.
General comments:
-Lines 108-109: What is the difference between these data bases?
-Lines 112-115: What could be the cause of this difference depending on the database used? Because the score obtained by AHTpin for SWMHQPP is quite low, while by BIOPEP it is the second peptide with antihypertensive potential. Also by mAHTPred the score is lower than expected by BIOPEP.
-Lines 124-126: What is the purpose of evaluating the IC50 values? To compare them with other peptides described on bibliography? At least the authors should describe which values they considered for ai and N.
-Lines 126-133 should be removed from this section and translated to section 2.2.1, after evaluating the IC50 values of the peptides by in vitro methodologies.
-Lines 133-136: Antihypertensive peptides have been reported from many natural sources, which is the point of mentioning these in particular?
-Lines 138-141 and 147-148: I consider that it is not appropriate to make this classification in such arbitrary way. The discussion should consider the actual doses used for common antihypertensive drugs, in order to evaluate if an IC50 value in the range of 50 µg/mL is more or less effective than actually pharmaceuticals.
-Lines 150-153: Propose some hypothesis for this observation.
-Table 5: iACE on digested peptides should be presented with the numbers and deviations, not with “>50”. “n.an” should be defined. SD is a mean of how many replicates?
-Line 165: These are the 3 peptides with high potential to be hypertensive according to BIOPEP, not by AHTpin and mAHTPred. Why the authors include SWMHQPP on the top 3 peptides while on the other bases the probability is lower than other peptides?
-Line 171: Moderate comparing to which peptides? What is the value of antioxidant activity to consider that a peptide presents low/moderate/high antioxidant potential.
-Lines 183-185: If the authors include each number and deviation it could be discussed the modification of the IC50 with the digestion process.
-The authors included in vitro antioxidant activity but did not perform any discussion of the in silico analysis on this activity.
I consider that the paper could be interesting if the authors perform a more extensive analysis on the in silico study, and compare the results with more in vitro biological activities, such as DPP-IV inhibitory activity, alpha-glucosidase, etc.
Author Response
The paper by Coscueta and co-authors entitled “Screening of novel bioactive peptides from goat casein: in silico to in vitro validation” compares in silico results for bioactive peptides with in vitro analyses on antihypertensive and antioxidant activity. Although the topic of the manuscript is interesting it has not been well conducted. Extensive editing of English language and style is also required. In my opinion, the paper is not suitable for publication in the prestigious International Journal of Molecular Sciences.
R: We appreciate the recommendation on the English, however, the reviewer makes the general comment on the need for extensive editing without giving any examples. It is worth mentioning that the article was reviewed prior to submission by a native speaker as well as the other reviewer highlighted the correct English.
General comments:
-Lines 108-109: What is the difference between these data bases?
R: In section "4.2.1. Bioactivity prediction" the tools used are specified. We do not provide in-depth detail of how each one works, because this is a topic more related to computer engineering, which is beyond the scope of this work, where we only evaluate the tools available to those of us who work in the analysis and production of bioactive compounds.
-Lines 112-115: What could be the cause of this difference depending on the database used? Because the score obtained by AHTpin for SWMHQPP is quite low, while by BIOPEP it is the second peptide with antihypertensive potential. Also by mAHTPred the score is lower than expected by BIOPEP.
R: Again, this is related to the code behind each tool, which is not the analysis of this work. In lines 20 and 21 we state "This study aims to draw attention through a critical study on how bioinformatics analysis is employed in the identification of bioactive peptides in the laboratory", Thus, it is not the aim of our work to understand the bioinformatics assumptions of the programmes.
-Lines 124-126: What is the purpose of evaluating the IC50 values? To compare them with other peptides described on bibliography? At least the authors should describe which values they considered for ai and N.
R: Sorry, but nothing is said about IC50 values in those lines. If what the reviewer is referring to is what is expressed in lines 148-151 (140-143 in the version before the revisions), the IC50 calculation is to compare with what is reported for other pure peptides, as that is the common way in iACE analysis. Each term in the equation of B is specified on lines 286-290 (273-277 in the version before the revisions). The sequence of each peptide is shown, so it is not necessary to specify the N (which is well specified to be the number of amino acids in the analyzed sequence), the same for ai, since if we are talking about the peptide as a whole and not a specific sequence that repeats within the peptide, ai acquires value 1.
-Lines 126-133 should be removed from this section and translated to section 2.2.1, after evaluating the IC50 values of the peptides by in vitro methodologies.
R: Again, the quoted lines do not correspond to the description. If the reviewer is referring to lines 152-158 (144-150 in the version before the revisions), we appreciate the suggestion, likewise the article tells a story that begins with an in silico analysis, and that paragraph is part of that analysis prior to the in vitro validation analysis.
-Lines 133-136: Antihypertensive peptides have been reported from many natural sources, which is the point of mentioning these in particular?
R: Once again, the quoted lines do not correspond to the description. In this case we could not identify what the reviewer means by "these in particular".
-Lines 138-141 and 147-148: I consider that it is not appropriate to make this classification in such arbitrary way. The discussion should consider the actual doses used for common antihypertensive drugs, in order to evaluate if an IC50 value in the range of 50 µg/mL is more or less effective than actually pharmaceuticals.
R: Again, the lines cited do not correspond to the description. We work with natural compounds and focus the work on comparing with other equivalent natural compounds, that is, other peptides, which is what the bioinformatics tools analyzed do, and not on comparing with commercial drugs. It is an interesting proposal, but it is not what the proposed work is looking for.
-Lines 150-153: Propose some hypothesis for this observation.
R: Again, the lines cited do not correspond to the description. In this case we are unable to identify which observation the reviewer is referring to.
-Table 5: iACE on digested peptides should be presented with the numbers and deviations, not with “>50”. “n.an” should be defined. SD is a mean of how many replicates?
R: As expressed in lines 200-202 (187-189 in the version before the revisions) "We performed the digestions considering reaching a concentration of 50 µg mL-1 in the simulation of the intestinal compartment", so if inhibition is not achieved at that concentration, there is no mean value with deviation to report, simply that the IC50 is greater than that concentration.
-Line 165: These are the 3 peptides with high potential to be hypertensive according to BIOPEP, not by AHTpin and mAHTPred. Why the authors include SWMHQPP on the top 3 peptides while on the other bases the probability is lower than other peptides?
R: Once again, the lines quoted do not correspond to the description. SWMHQPP is not only considered by BIOPEP (although this is the main tool used, on which the others are also based), but also by PeptideRanker. This choice was then validated in vitro as it presented the highest inhibitory potential.
-Line 171: Moderate comparing to which peptides? What is the value of antioxidant activity to consider that a peptide presents low/moderate/high antioxidant potential.
R: Once again, the lines quoted do not correspond to the description. When such mention is made, reference is made to well-established works in the literature, which we mention again below:
12. Power, O.; Jakeman, P.; Fitzgerald, R.J. Antioxidative peptides: Enzymatic production, in vitro and in vivo antioxidant activity and potential applications of milk-derived antioxidative peptides. Amino Acids 2013, 44, 797–820, doi:10.1007/s00726-012-1393-9.
13. Contreras, M. del M.; Hernández-Ledesma, B.; Amigo, L.; Martín-Álvarez, P.J.; Recio, I. Production of antioxidant hydrolyzates from a whey protein concentrate with thermolysin: Optimization by response surface methodology. LWT - Food Sci. Technol. 2011, 44, 9–15, doi:10.1016/j.lwt.2010.06.017
-Lines 183-185: If the authors include each number and deviation it could be discussed the modification of the IC50 with the digestion process.
R: Again, the lines cited do not correspond to the description. The answer is the same to the question posed in Table 5.
-The authors included in vitro antioxidant activity but did not perform any discussion of the in silico analysis on this activity.
R: The proposal is interesting; the problem is that the bioinformatics tools analyzed are generally for inhibitory activities of proteases such as ACE. For this type of activities, values such as B are presented, which is not the case for antioxidant activity.
I consider that the paper could be interesting if the authors perform a more extensive analysis on the in silico study, and compare the results with more in vitro biological activities, such as DPP-IV inhibitory activity, alpha-glucosidase, etc.
R: We appreciate the comment, and we also see interest in the idea. On the other hand, the reviewer should not lose sight of the typology of the article, it is a short communication and not a full article.
Round 2
Reviewer 2 Report
I appreciate the comments of the authors, who have clarified my doubts.